# Species-wide survey of the expressivity and complexity spectrum of traits in yeast

Andreas Tsouris[1], Téo Fournier[1], Anne Friedrich[1], Jing Hou[1], Maitreya J. Dunham[2], Joseph Schacherer[1,3]*

1 Université de Strasbourg, CNRS, GMGM UMR 7156, Strasbourg, France, 2 Genome Sciences Department, University of Washington, Seattle, Washington, United States of America, 3 Institut Universitaire de France (IUF), Paris, France

* schacherer@unistra.fr

**Data Availability Statement:** Phenotypic data are available online at http://1002genomes.u-strasbg.fr/files/Diallel_Phenotypes/«.

**Funding:** This work was supported by a National Institutes of Health (NIH) grant R01 (GM147040-

## Abstract

Assessing the complexity and expressivity of traits at the species level is an essential first step to better dissect the genotype-phenotype relationship. As trait complexity behaves dynamically, the classic dichotomy between monogenic and complex traits is too simplistic. However, no systematic assessment of this complexity spectrum has been carried out on a population scale to date. In this context, we generated a large diallel hybrid panel composed of 190 unique hybrids coming from 20 natural isolates representative of the *S. cerevisiae* genetic diversity. For each of these hybrids, a large progeny of 160 individuals was obtained, leading to a total of 30,400 offspring individuals. Their mitotic growth was evaluated on 38 conditions inducing various cellular stresses. We developed a classification algorithm to analyze the phenotypic distributions of offspring and assess the trait complexity. We clearly found that traits are mainly complex at the population level. On average, we found that 91.2% of cross/trait combinations exhibit high complexity, while monogenic and oligogenic cases accounted for only 4.1% and 4.7%, respectively. However, the complexity spectrum is very dynamic, trait specific and tightly related to genetic backgrounds. Overall, our study provided greater insight into trait complexity as well as the underlying genetic basis of its spectrum in a natural population.

## Author summary

Dissecting the genetic origins of natural phenotypic variation is a major goal in biology. In 1865, Gregor Mendel established principles of inheritance that described the transmission of genetic traits. However, we still lack a precise view of the spectrum and continuum of trait complexity in natural population. In this context, we carried out a study of the complexity of traits in a large population of isolates using the yeast *Saccharomyces cerevisiae*. We analyzed patterns of distribution and inheritance of offspring of a wide diallel panel and in a large number of environments. We found that on average 91.2% of the traits are complex, while only 4.1% and 4.7% are monogenic and oligogenic, respectively. However, it is also clear that the complexity spectrum depends on genetic background and environment. Interestingly, we have highlighted and dissected the genetic basis of

01) to M.J.D and J.S., as well as a European Research Council (ERC) Consolidator grant (772505) to J.S. It is also part of Interdisciplinary Thematic Institutes (ITI) Integrative Molecular and Cellular Biology (IMCBio), as part of the ITI 2021-to-2028 program of the University of Strasbourg, CNRS, and Inserm, supported by IdEx Unistra (ANR-10-IDEX-0002). The funders had no role in study design, data collection and analysis, decision to publish, or preparation of the manuscript.

**Competing interests:** The authors have declared that no competing interests exist.

cases showing a broad complexity spectrum, such as in the presence of copper sulfate as well as galactose as a carbon source.

## Introduction

The independent rediscovery of Mendel's laws in the early 1900s by De Vries, Correns and Tschermak has been a keystone for modern genetics [1–3]. However, the possibility of having higher complexity in the inheritance of traits was quickly highlighted [4]. In 1920, Altenberg and Muller first dissected a complex trait, the deformation of the wing shape in *Drosophila melanogaster* [5]. It has since become abundantly clear that there is a broad spectrum and continuum existing between Mendelian and complex traits within any natural population. There is growing evidence that monogenic mutations do not always strictly adhere to Mendelian inheritance, with a hidden complexity behind some cases. It has been shown in both model organisms and human genetic studies that the effect of a given variant can be highly variable across multiple genetic backgrounds and can be modulated by the combined action of other variants [6–16].

Several large-scale surveys on different model organisms, such as the yeast *Saccharomyces cerevisiae*, have thus highlighted the broad influence of genetic backgrounds on the phenotypic landscape. Nevertheless, we still lack a comprehensive view of the dynamics of trait complexity spectrum at the population level. In fact, the underlying genetic complexity of a trait can be assessed by examining patterns of inheritance. Ideally, population-level exploration would analyze progeny resulting from various crosses of a large number of genetically different parental strains, *i.e.* using a diallel design [17–19]. In this context, the yeast *S. cerevisiae* is a powerful model as natural populations of isolates from various environments (*e.g.*, including soil, tree barks, different insects, immunodepressed patients) and fermentation processes (*e.g.*, wine, bread, and bioethanol), exhibit a high genetic diversity [20–27]. In addition, isolates can be crossed with each other to give a large progeny and analyzing the phenotypic distribution of segregants makes it possible to evaluate the genetic complexity of the traits. In addition, a unique and particularly attractive feature of these *S. cerevisiae* populations is that all segregants can be selected from full tetrads, allowing the complete genetic information from any meiotic event to be preserved. Such a design provides the possibility of accessing the complexity and heritability of any trait of interest by analyzing its distribution and expressivity at the population level.

By crossing a single *S. cerevisiae* lab strain (namely, Σ1278b) to 41 natural isolates and phenotyping their offspring on 30 growth conditions impacting various cellular pathways, a first estimation of the monogenic compared to complex inheritance has been previously carried out [13]. This study also showed the dynamic of trait complexity depending on the genetic backgrounds that a particular variant lies in. Indeed, an isolate containing a variant conferring resistance to cycloheximide and anisomycin was crossed with 20 isolates sensitive to these compounds. Offspring analysis showed that in 30% of the cross, a deviation from a Mendelian inheritance was observed [13]. This expressivity reflected the presence of genetic modifiers in some of the explored genetic backgrounds. However, this study suffered from several biases. First, with respect to estimating the prevalence of Mendelian inheritance, the full extent of genetic diversity has not been explored because a single strain has been consistently crossed with many. Moreover, strong allelic effects that are specific to a particular background might impact several crosses in a similar way thus inducing a bias. Finally, a diallel design could highlight certain specific complex cases linked to precise parental combinations and consequently

the broad spectrum of expressivity in a population. Extending this study by performing a "many by many" cross instead of a "one by many" is therefore essential to obtain a systematic and unbiased view of the genetic complexity of traits as well as measuring expressivity for variants with important phenotypic effect.

Here, we combined the power of classical yeast genetic techniques with high throughput phenotyping and machine learning algorithms to get the first species-wide view of genetic complexity of traits but also to investigate expressivity through the lens of genetic complexity in a high number of cross/trait combinations. Twenty *S. cerevisiae* natural isolates that are representative of the entire species diversity were crossed in a pairwise manner to obtain 190 unique hybrids. Then we obtained a large progeny of 160 individuals for each of these crosses leading to 30,400 individuals. The phenotyping of this diallel offspring panel on 38 growth conditions impacting different physiological pathways allowed us to analyze the phenotypic distribution and segregation patterns of the progenies. Using a classification algorithm that we developed, it was possible to evaluate the level of complexity of 6,870 crossover/trait combinations and we found that on average 91.2% of cases are complex, with a variable fraction ranging from 46.4% to 99% depending on conditions. On average, monogenic and oligogenic cases accounted for only 4.1% (with a range between 0.5% and 48.2%) and 4.7% (with a range between 0.6% and 18.6%), respectively. The complexity spectrum is clearly variable across traits and genetic backgrounds. Examination of the phenotypic distribution in a panel of diallel offspring also provided clues to the genetic basis affecting monogenic and oligogenic cases, and hence the complexity spectrum.

## Results

### Generation of a diallel offspring panel

In principle, the genetic underpinnings of the parental strains are essentially reshuffled in their offspring. Therefore, analyzing the phenotypic distribution in the progeny allows easy assessment of the genetic complexity of these traits. In yeast, the genetic complexity of a trait can be inferred from the phenotypic distribution and segregation obtained in the haploid progeny of a cross, with 3 categories according to complexity, namely monogenic, oligogenic and complex traits (Fig 1A). For a Mendelian trait, the contrasting phenotype between the parental isolates is controlled by a single locus, so half of the offspring inherit the causal allele and 2:2 segregation in any given tetrad is observed in yeast (Fig 1A). Therefore, the global offspring growth distribution follows a bimodal pattern with equal partitioning of segregants in either parental phenotype group. Oligogenic traits, usually influenced by a few genes, represent an intermediate between Mendelian and complex traits. In our design, oligogenic traits exhibit bimodal phenotypic distributions, but there is a clear deviation from 2:2 Mendelian segregation and an uneven repartition of segregants in either parental phenotype is observed (Fig 1A). Finally, complex traits produce unimodal phenotypic distributions in which most individuals have an average phenotype, whereas fewer individuals have extreme phenotypes (Fig 1A).

To examine the spectrum of genetic complexity characteristic of a natural population, we first generated a half-diallel panel with stable haploid lines of 20 natural *S. cerevisiae* isolates (Fig 1B). To cover a large part of the genetic and phenotypic diversity of the species, parental strains were selected from many different ecological niches (*e.g.*, wine, soil, clinical) and various geographical locations (*e.g.*, France, China, South Africa, and Ecuador) (S1 Table). The average nucleotide diversity between two parents ranges from 0.56% to 1.05% (S2 Table). The 20 selected isolates were crossed in an all by all manner without reciprocal crosses or homozygous crosses leading to a half diallel cross of 190 hybrids (Fig 1B). For each of these hybrids, a

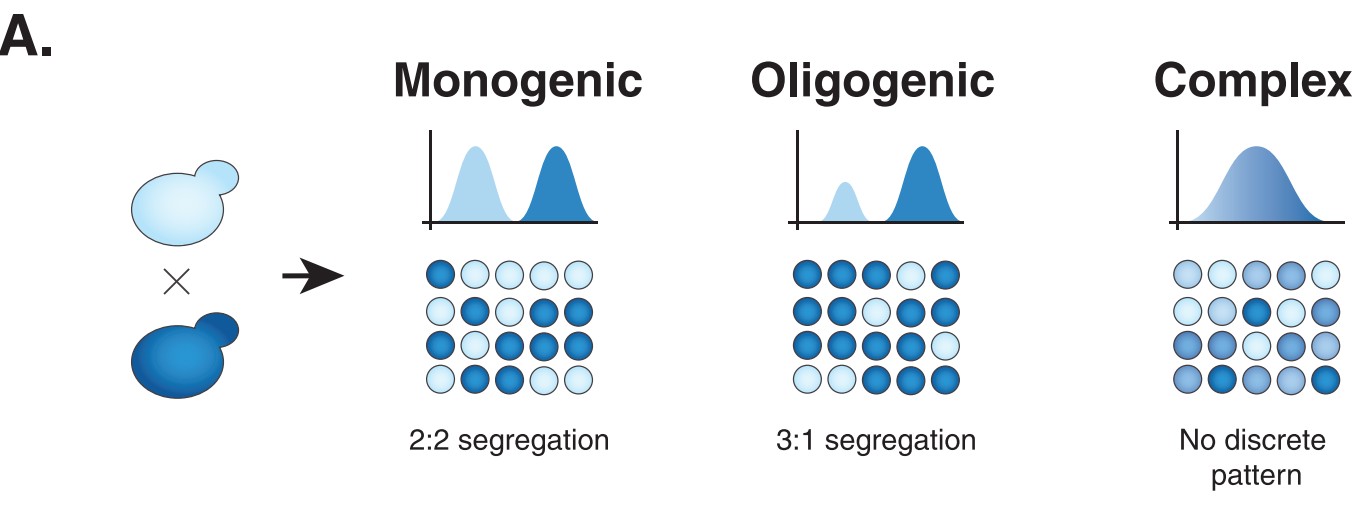

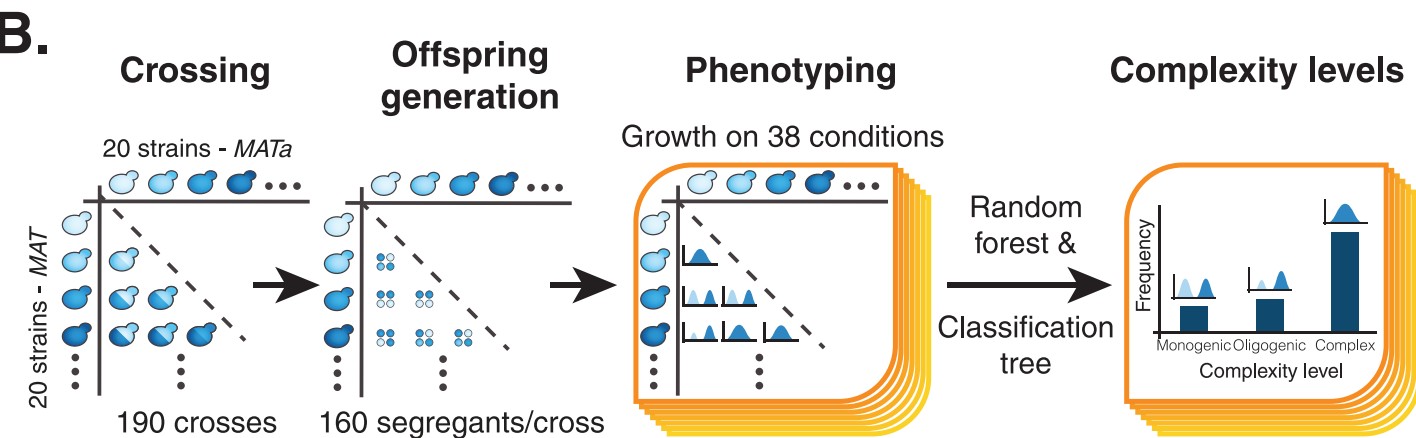

**Fig 1. Genetic complexity and phenotypic distribution of yeast segregant populations.** (A) Populations of yeast segregants have specific phenotype distributions and segregation patterns depending on the complexity of the trait. Monogenic traits produce bimodal distributions, and each tetrad contains 2 individuals from each phenotype group, also called a 2:2 segregation. Oligogenic traits also produce bimodal distributions but don't necessarily follow a 2:2 segregation pattern as they can also have a 3:1 and 0:4 segregation. Lastly, complex traits produce unimodal phenotypic distributions that resemble the shape of the normal distribution. (B) Here, we established a half diallel cross between 20 parental strains which generated 190 unique crosses. We then generated a segregant population of 160 individuals (40 full tetrads) for each cross and measured the growth of all the segregants and parents on 38 conditions. We studied the phenotypic distributions of all crosses and predicted the genetic complexity of growth on each condition for every one of the 190 crosses.

large progeny of 160 haploids coming from 40 tetrads with four viable spores were obtained, summing up to 30,400 spores.

## Inferring the complexity from the phenotypic distributions

To infer the level of trait complexity and assess its dynamics in the phenotypic landscape, we first performed phenotyping of the entire panel of 30,400 haploid progeny coming from the 190 hybrids. We selected 38 conditions impacting various cellular pathways (S3 Table) and measured their mitotic growth ability on solid media by assessing colony sizes (See Methods). From more than two million phenotypic measurements grouped for each cross and condition (trait), we obtained approximately 6,870 phenotypic distributions of haploid progenies, *i.e.* one distribution for each cross/trait combination. The inheritance pattern reflects the genetic complexity of a trait in a given cross between two specific genetic backgrounds.

Due to the large number of phenotypic distributions, we developed a classification algorithm that can classify the phenotypic distribution in the three complexity levels (Fig 2). Our classification algorithm is a combination of a random forest followed by a decision tree that takes into account the phenotypic distribution, the segregation in each tetrad and the parental phenotypes. We first predicted whether the distribution is bimodal or unimodal using a random forest model that assesses the shape of the distribution and the segregation of the phenotypes in the tetrads (Fig 2A and 2B) (See Methods). This model was trained with 50,000 simulated sets of 160 phenotypes segregating in 40 tetrads, similar to our experimental phenotypes (See Methods). The model was then evaluated on a set of 545 manually annotated real phenotypic distributions and gave an AUC (area under the curve) score of 0.977 (Fig 2C). The bimodality predictions of the random forest model and the phenotypes of the parents were then passed to a decision tree that classified the cross/trait as monogenic, oligogenic or complex (S1 Fig). Overall, we found that most cross/trait combinations (ranging from 46.4% to

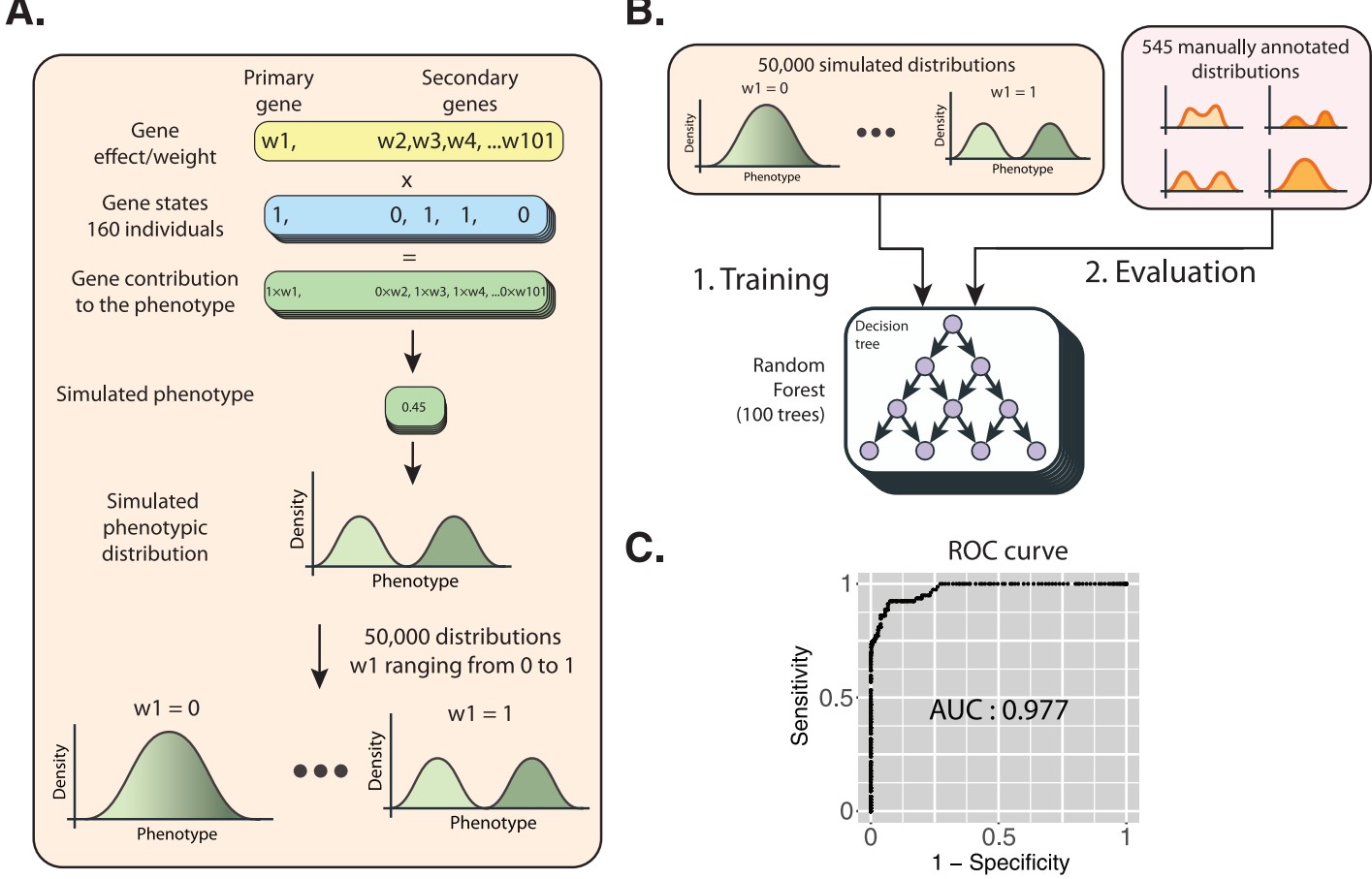

**Fig 2. Methods used for the classification of genetic complexity.** (A) Pipeline of the generation of simulated phenotypic distributions using a non-epistatic model. We first defined 101 "genes" and assigned them binary states (0/1) that follow the segregation of biallelic variants in the cross between two parents with a genotype of 0 and 1, respectively. Each "gene" was assigned a random weight and the phenotype of each individual was defined as the sum of the products between the "gene" weights and their binary states. The phenotypes were calculated for 160 individuals segregating in 40 tetrads, as our diallel populations and these simulated phenotypes were used to generate a phenotypic distribution. We generated 25,000 phenotypic distributions with this model and 25,000 using two epistatic models for a total of 50,000 distributions at different levels of bimodality. (B) The 50,000 simulated distributions were used to train a random forest model (of 100 trees) that predicts the weight of the primary "gene". This model was evaluated using a set of 545 manually annotated distributions from our phenotyping experiment and (C) produced a receiver operating statistic curve (ROC) with an area under the curve (AUC) of 0.977. To classify distribution as bimodal or unimodal from the predicted "gene" weight we used a threshold that ensures an equal compromise between sensitivity and specificity.

99% of the cases depending on conditions) exhibit complex patterns. Oligogenic and monogenic cases are less frequent and represent, depending on the conditions, between 0.5% and 48.2%, and between 0.6% and 18.6%, respectively (Fig 3A and S4 Table). However, complexity spectrum is highly variable depending on the conditions. Although only complex cases are observed in most conditions (*e.g.*, SC formamide 5%, SC glycerol 10%), only a few conditions show a variable spectrum in the population (*e.g.*, SC $CuSO_4$ 1 mM, SC SDS 0.01%, SC galactose 2%) (Fig 3B). Overall, about 90% of the conditions show less than 20% of the crosses exhibiting low complexity cases.

To quantify the magnitude of the complexity spectrum across the crosses of each parent under the 38 different conditions, we then determined the Shannon entropy (Fig 3C and S5 Table). This statistic is used to quantify the uncertainty inherent in the possible outcomes of a variable, which in our case corresponds to complexity. High entropy values indicate that complexity is highly variable, whereas low entropy values indicate that the complexity is constant. As expected, conditions with a high number of low complexity cases (*i.e.*, SC $CuSO_4$ 1 mM, SC SDS 0.01%, SC galactose 2%) exhibit a high entropy score for most parental isolates (Fig 3C). Entropy scores are also variable depending on parental lines (Fig 3C).

Altogether, these results clearly highlight a variability of the complexity spectrum depending on the conditions. In addition, the complexity of inheritance really depends on the parental cross and is not intrinsic to a trait.

## High variability in complexity spectrum in the presence of $CuSO_4$

One of the main advantages of using a diallel design is that we can track the effect of a causative genetic variant in multiple genetic backgrounds distributed across the genetic diversity of the whole population. By examining the genetic complexity of all phenotypic distributions sharing a parent, we can detect the presence of major loci with high phenotypic impact. Such a variant is expected to lead primarily to monogenic inheritance in the offspring of each cross involving that particular strain. In addition, deviation of this Mendelian inheritance will lead to a larger complexity spectrum as observed in some conditions such as SC $CuSO_4$ 1 mM and SC galactose 2%. We therefore sought to further explore the genetic cause of this spectrum within these two conditions.

Regarding the growth and resistance in presence of copper sulfate ($CuSO_4$), it has recently been shown that such trait follows a bimodal distribution model [13], and therefore a Mendelian inheritance pattern in a large population of 1,011 natural isolates [24]. Genome-wide association study performed on this set of isolates highlighted the main locus corresponding to the *CUP1* gene, which encodes for a copper binding metallothionein. In fact, amplification of this gene strongly contributes to the resistance to high concentrations of copper and cadmium [28] with copy number variation alone explaining 44.5% of phenotypic variation [24].

We therefore looked at the involvement of the *CUP1* gene in the variable complexity observed in our diallel offspring panel. First, we observed a clear correlation between growth in copper sulfate and *CUP1* copy number for parental isolates (Fig 4A). Second, parents with a single *CUP1* copy cause more monogenic and oligogenic cases than parents with multiple copies (Fig 4B). Finally, we clearly observed that all monogenic cases systematically arise from crosses between parental isolates with one *CUP1* copy and parents with multiple *CUP1* copies (Fig 4C). In contrast, crosses between parents with a similar copy number most often result in complex distributions (Fig 4C). It is also important to note that the level of complexity has obviously not been determined for a number of crosses without growth in this condition, often crosses between two parents with a single copy of *CUP1*. Overall, these results clearly

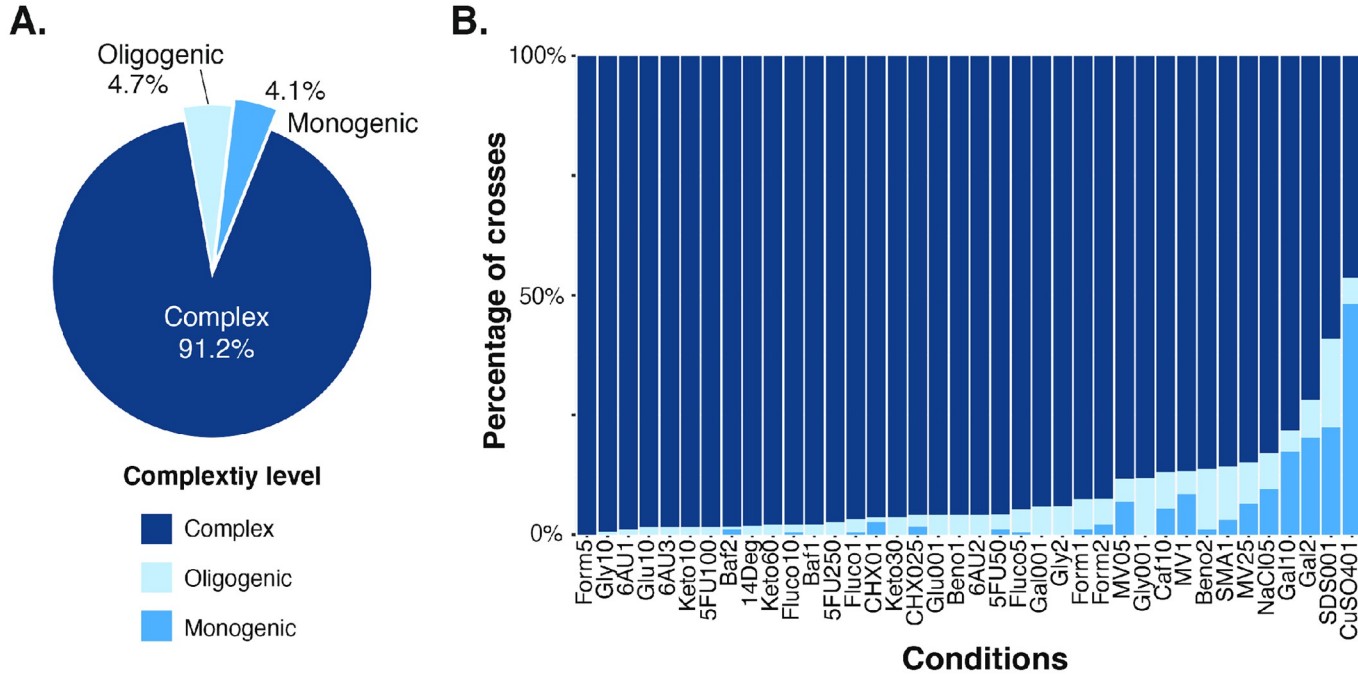

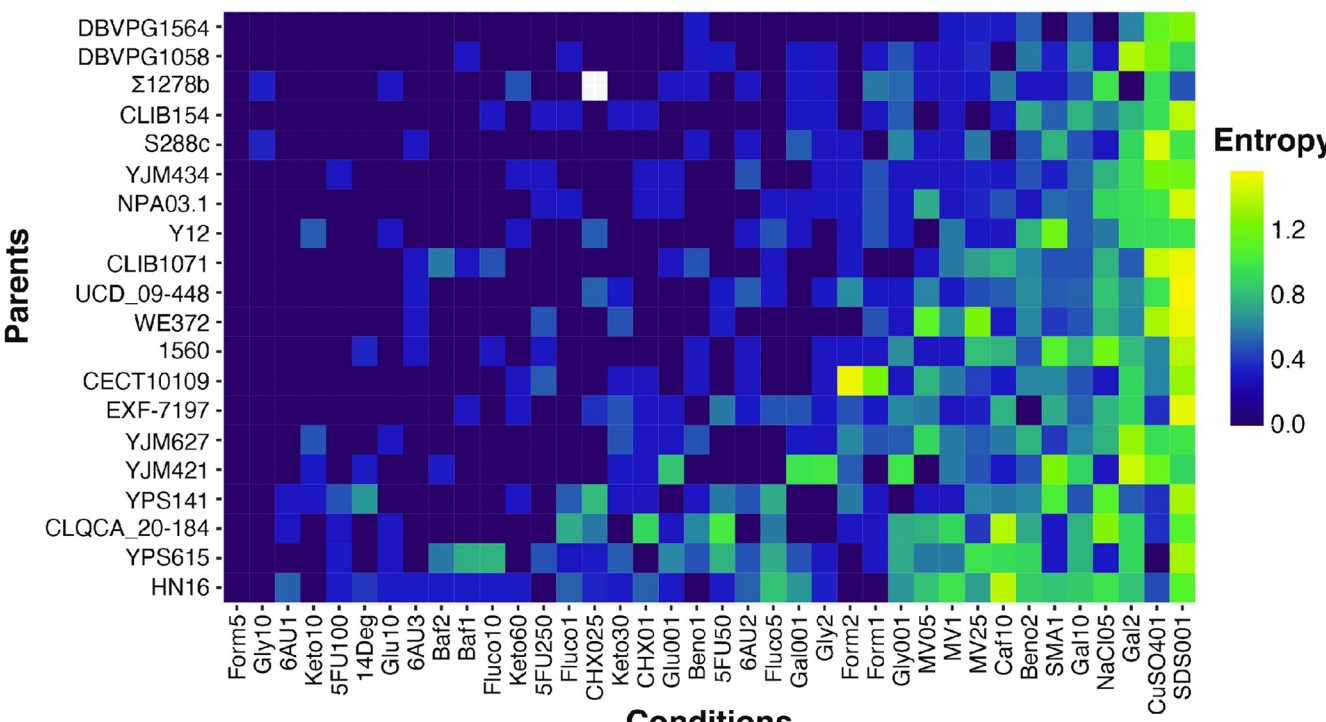

**Fig 3. The genetic complexity of 38 growth traits over 190 genetic backgrounds.** (A) The genetic complexity of every cross/growth trait combination was predicted. Overall, most cross/trait combinations (91.2%) were classified as complex traits (dark blue) while 4.7% are oligogenic (light blue) and 4.1% were classified as monogenic (blue). (B) The genetic complexity varies considerably across the 38 growth traits (conditions). (C) Comparison of the Shannon's entropy values for the crosses of each parent across different growth traits. The traits are sorted from the lowest to the highest average entropy value, from left to right.

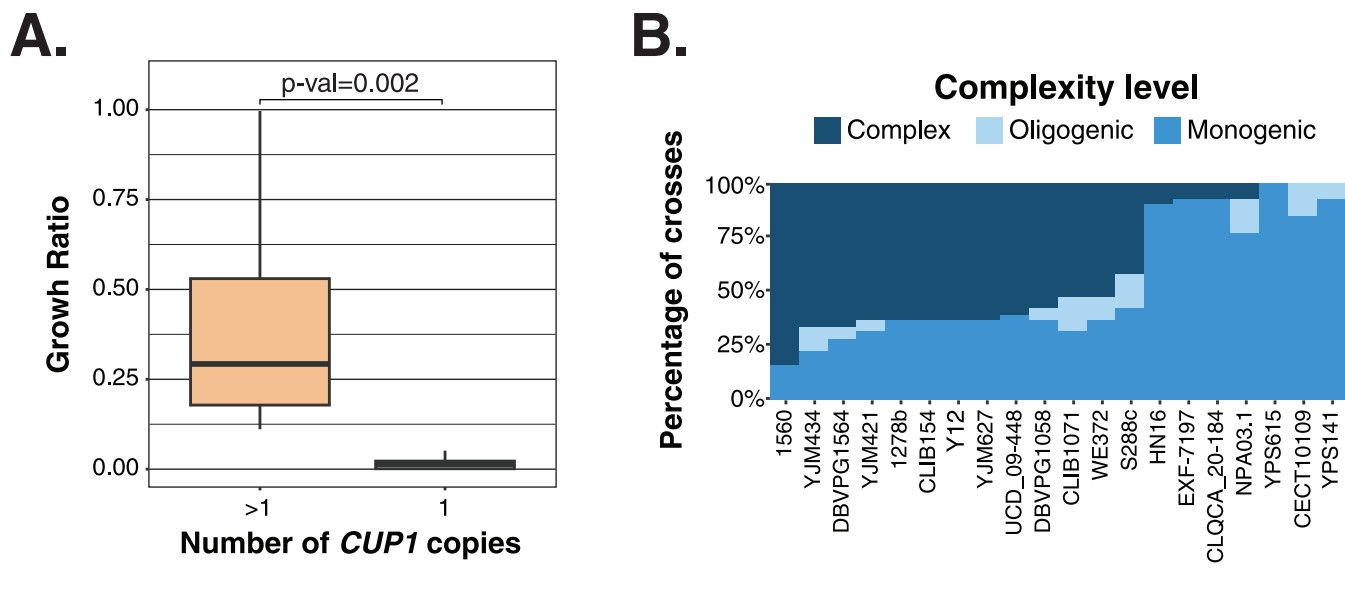

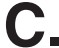

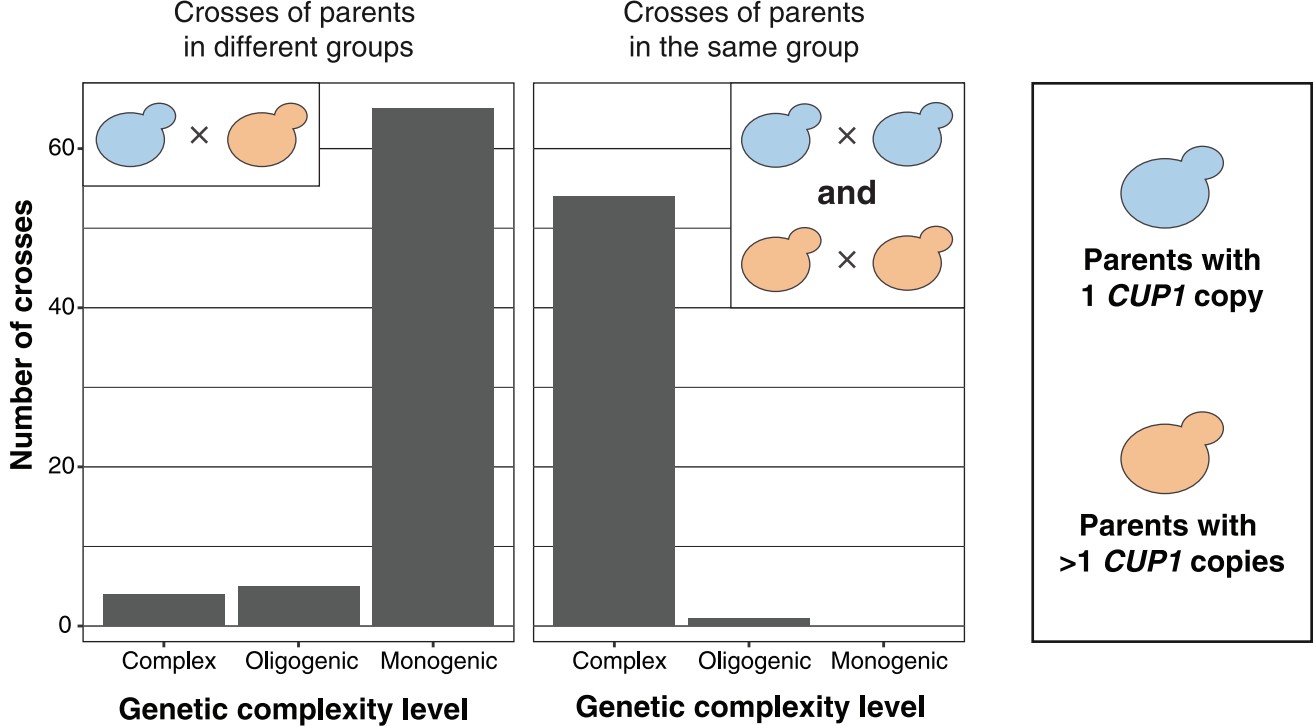

**Fig 4. Impact of *CUP1* copy number on growth in CuSO₄ 0.1mM.** (A) Comparison of the growth ratio of parental strains with more than 1 *CUP1* copy and parental strains with only one *CUP1* copy (two-sided t-test, p-value = 0.002). (B) Percentage of the CuSO₄ 0.1mM growth complexity levels across the crosses of the 20 parental strains. (C) Comparison of the complexity levels between the crosses of parents from different copy number groups and parents in the same copy number group. The two copy number groups are defined as the parental strains with only one *CUP1* copy (blue) and the strains with more than one *CUP1* copy (orange).

show that the complexity spectrum is driven by a main locus and in this specific example by the *CUP1* gene.

## Genetic basis of galactose complexity spectrum

Significant variation in genetic complexity was also observed for growth on galactose 2% with many genetic backgrounds having monogenic and oligogenic inheritance (Fig 3B). This deviation towards low complexity was most pronounced in the crosses involving four parents (NPA03.1, YJM627, YJM421 and DBVPG1058), where growth follows monogenic or oligogenic inheritance in more than half of their crosses (Fig 5A). This monogenic/oligogenic inheritance was always caused by a group of segregants with low growth, a phenotype similar to that of the four parents mentioned above (Fig 5B). All four parents have significantly lower growth in galactose than the rest of the parental strains (S2 Fig), which together with the prevalence of low complexity in their crosses suggests that these parents carry individual variants that greatly decrease growth on galactose 2%.

We therefore performed bulk-segregant analysis (BSA) followed by genome sequencing to pinpoint the loci with large effect on this phenotype [13,29,30]. To identify the variant in YJM627, we focused on the cross between YJM627 and YPS141 that follows monogenic inheritance (Fig 5B). We generated a population of approximately 200 segregants and screened for their growth on galactose 2%. We then generated two pools, one containing 100 low-growth segregants and another with high-growth 100 segregants. The two pools were sequenced in order to infer the frequency of the parental alleles in each of them. A significant deviation in allele frequency was observed in a region of approximately 100 kb on chromosome 4 (coordinates 400,000–500,000) (S3 Fig). This region contains *GAL3*, a gene coding for a transcription factor responsible for the expression of many genes involved in the catabolism of galactose [31]. Interestingly, the *GAL3* allele of YJM627 has a nonsense variant (*GAL3*, C456A), which lead to a non-functional version of *GAL3*. To test whether this version of *GAL3* is indeed the causal locus leading to low growth in galactose of YJM627 and the segregants carrying it, we introduced a centromeric plasmid with a functional allele of *GAL3* in the YJM627 strain [32]. Plasmid introduction led to high growth in galactose, confirming the role of *GAL3* and more particularly its loss of function in the observed phenotypic variation (Fig 5C). Regarding the case of the parental strain NPA03.1, we found that the cross between YJM627 and NPA03.1 only produces segregants with low growth values, which is most likely due to the fact that the causal locus is genetically linked in these two strains (S4 Fig). The large-effect variant in NPA03.1 is therefore most likely present in *GAL3*, as it was in YJM627. Similar to what was done for YJM627, we introduced the same centromeric plasmid with a functional *GAL3* allele in NPA03.1, which dramatically increased growth, verifying that *GAL3* is responsible for the decreased growth in NPA03.1 and the low complexity of many of its crosses (Fig 5C).

We then focused on the DBVPG1058 parent, which is also involved in many crosses following monogenic inheritance (Fig 5A). We used the cross between DBVPG1058 and YJM627 to perform bulk segregant analysis. Both parents have high-effect variants leading to low growth on galactose, and offspring show oligogenic inheritance (Fig 5B). We performed BSA with 2 pools of 100 segregants showing low and high growth on galactose, respectively. Low-growth segregants had large allele frequency deviations for two regions, one on chromosome 4 (coordinates 400,000–500,000) and one on chromosome 12 (coordinates 250,000–350,000) (S5 Fig). The region of chromosome 4 corresponds to the *GAL3* locus and the location of the large-effect variant in the YJM627 parental strain, the region of chromosome 12 corresponds therefore to the causal locus in the DBVPG1058 strain. This region of chromosome 12 contains

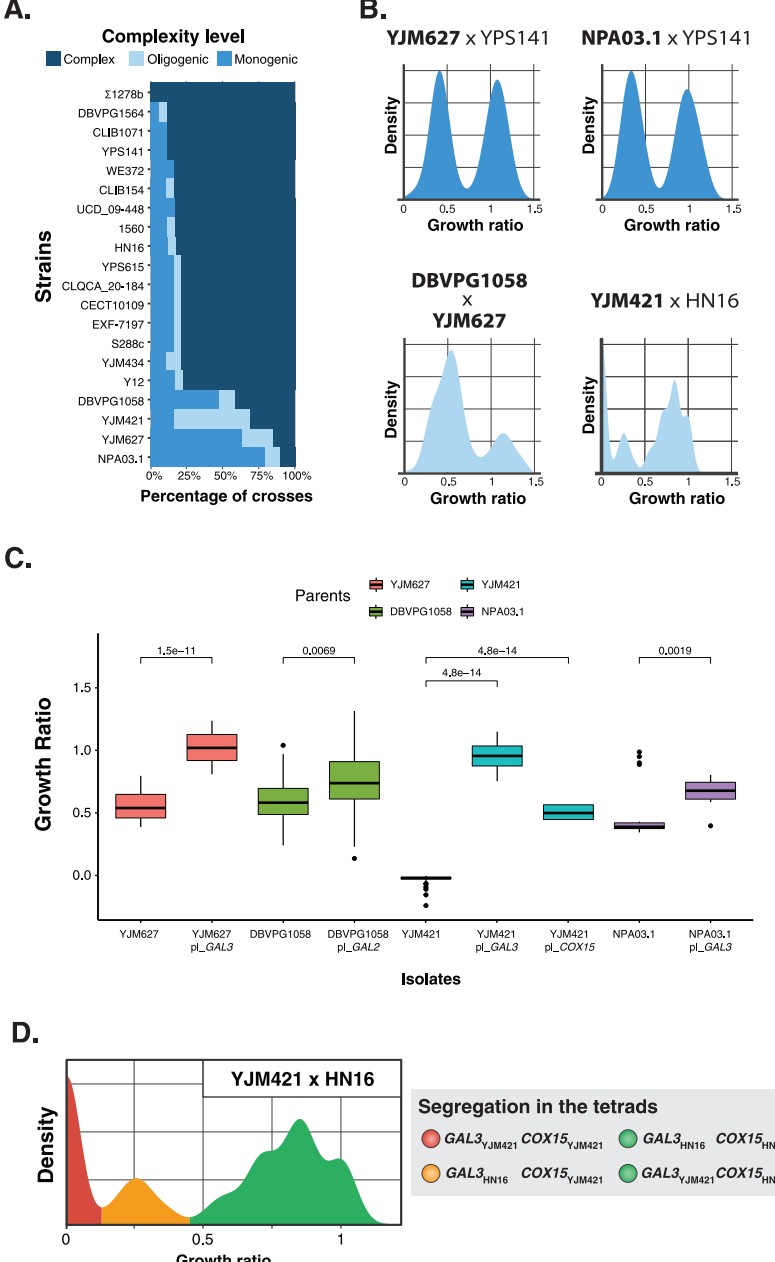

**Fig 5. Phenotypes of the parents with large effect variants in galactose.** (A) Percentage of crosses with different complexity levels in the crosses of each of each parent in galactose 2%. (B) Examples of phenotypic distribution that were classified as monogenic (blue) or oligogenic (light blue). The names of the 4 parents that mainly produce monogenic and oligogenic crosses are in bold while the names of the second parents in each cross are not in bold. (C). Phenotypic rescue of growth on galactose 2% by plasmid transformation. Comparison of the phenotypes of the 4 parents with large effect variants and the same parents after a transformation with centromeric plasmids containing functional versions of the candidate large effect genes (t-test between the non-transformed parent and its transformed counterparts). (D) The phenotypic distribution of the cross between YJM421 and HN16 shows 3 distinct groups of spores, no growth (red), low growth (orange) and high growth (green). These 3 groups follow a 1:1:2 segregation pattern in the tetrads (1 no growth spore, 1 low growth, 2 high growth spores) and are associated to specific alleles of the *COX15* and *GAL3* genes.

*GAL2*, the gene coding galactose permease responsible for importing galactose into the cell [33]. To verify that the *GAL2* allele of DBVPG1058 impacts growth on galactose, we transformed the parental strain with a centromeric plasmid containing a functional allele of *GAL2* [32]. Transformed cells exhibited higher growth on galactose than their untransformed counterparts, showing that monogenic/oligogenic inheritance in crosses involving DBVPG1058 are caused by its *GAL2* allele (Fig 5C).

The last parent leading to many crosses following monogenic or oligogenic inheritance is YJM421 (Fig 5A). Unlike other parents with high-effect variants, most low-complexity crosses of YJM421 follow oligogenic inheritance. We performed bulk segregant analysis on the cross between YJM421 and HN16 by forming two pools, a group containing segregants with low growth on galactose (n = 100) and another with segregants having high growth (n = 100). As expected, two regions showed a bias towards alleles of a specific parent. The first region is located on chromosome 4 (coordinates 350,000–500,000), the locus of *GAL3*, and the second region is located on chromosome 5 (coordinates 400,000–500,000) (S6 Fig). The region of chromosome 5 contains *COX15*, which was previously shown to be responsible for the decreased growth of YJM421 on glycerol 2% because of a nonsense variant at the position +115 in its open reading frame (C114T) [29]. Both candidate genes were validated by independently transforming YJM421 with two centromeric plasmids carrying functional versions of *GAL3* and *COX15*. In both cases, the strains containing the functional alleles of the genes grow better than the untransformed YJM421 strain, proving that the high-effect variants are located in *GAL3* and *COX15* (Fig 5C). To dissect the interaction between the two genes, we examined the genotypes of the segregants in each phenotypic group. This cross has three groups of segregants with distinct phenotypes (no, low and high growth) that follow a 1:1:2 segregation in each tetrad (Fig 5D). High-growth segregants carry the $COX15_{HN16}$ allele and the *GAL3* allele of either parent. Low-growth segregants have a $COX15_{YJM421}$ $GAL3_{HN16}$ genotype, while the segregants with no growth on galactose have $COX15_{YJM421}$ $GAL3_{YJM421}$ genotype. Therefore, *COX15* is the major locus, which controls growth as it is the differentiating factor between high-growth and growth-deficient (no/low growth) segregants. In deficient growth, *GAL3* can act as a modifier gene as the $GAL3_{HN16}$ allele can partially restore growth from no growth to poor growth.

Overall, we identified three genes (*GAL3*, *GAL2* and *COX15*) with large-effect variants in four parents. In the case of the $GAL3_{YJM627}$ and $COX15_{YJM421}$ alleles, the large-effect variants are identified nonsense variants. For the remaining cases, we do not know the precise location of the large-effect variants. To that end, we leveraged the genetic and phenotypic diversity of the 1,011 natural isolates collection [24]. We first identified the SNPs present in the *GAL3* and *GAL2* genes in the parental strains. For each of the SNPs identified, we tested whether the natural isolates carrying the same variant as the parent of interest (the low-growth parent) had significantly lower growth in galactose than those carrying the variant of the other parent (the high-growth parent) (one-sided t-test with Bonferroni normalization). We observed significantly lower growth for the isolates carrying the nonsense *GAL3* C456A variant present in strain YJM627, validating its effect on the phenotype (Fig 6). We were able to identify the large effect variants present in strains NPA03.1 and DBVPG1058 in the same way. The NPA03.1 variant is *GAL3* G154C, a non-synonymous variant converting a glycine residue to an arginine and the large-effect variant for DBVPG1058 is *GAL2* A655C, a non-synonymous variant converting a tryptophan residue to a proline (Fig 6).

Overall, we identified four SNPs with very large effects on growth in galactose present in four out of the 20 parents of the diallel cross (S6 Table). Interestingly, all of these large impact variants have a low frequency in the population of the 1,011 isolates making them very difficult to detect using a mapping strategy such as genome-wide association studies.

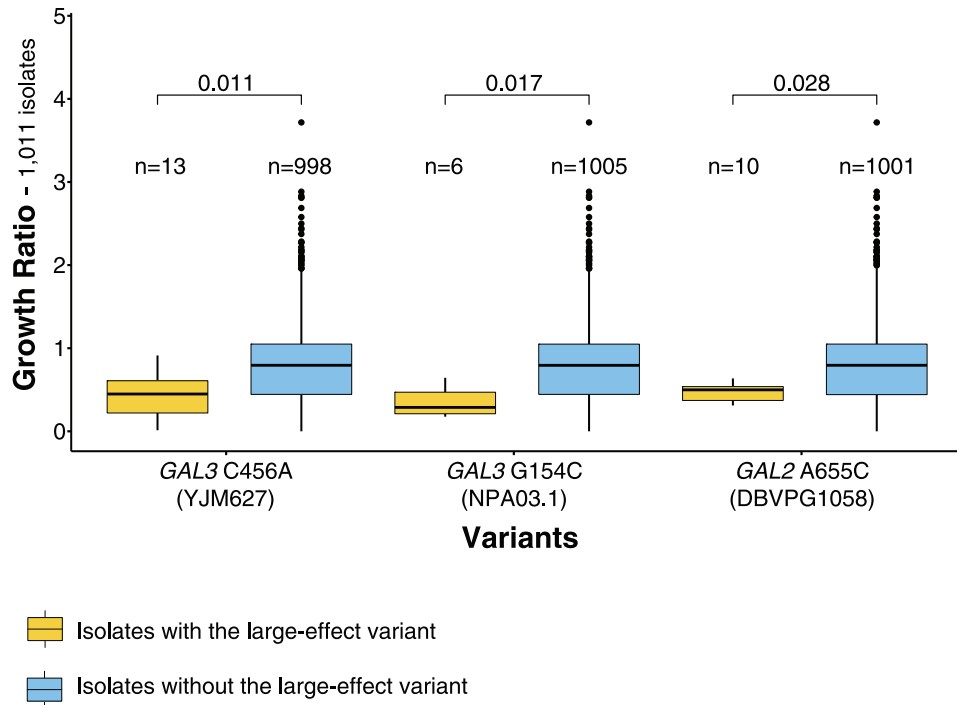

**Fig 6. Large effect variants and growth on galactose 2%.** Comparison of the growth on galactose 2% media between natural isolates of the 1,011 yeast isolates collection carrying the candidate large effect variants (yellow) and isolates carrying any other variant at that position (blue). The significance of the differences between the phenotypes of the two groups of isolates were assessed with a one-sided Wilcoxon test that tested whether the isolates with the variant had lower growth than the others. In cases with multiple candidate variants along the same gene, p-values were normalized using Bonferroni normalization.

## Discussion

By performing a species-wide screening of the genetic complexity of traits in *S. cerevisiae* with the progeny of 190 crosses from 20 natural isolates under 38 growth conditions, we were able to assess the complexity level of 6,870 cross/trait combinations. One of the main advantages of using segregating populations is the fact that we can determine the genetic complexity at population-scale from the phenotypic distribution. In the whole population, we found that on average 91.2% of cases are complex, and ranging from 46.4% to 99% depending on conditions. On average, monogenic and oligogenic cases accounted for only 4.1% (with a range between 0.5% and 48.2%) and 4.7% (with a range between 0.6% and 18.6%), respectively. The complexity spectrum is variable across traits as well as genetic backgrounds. Studying phenotypic distribution across crosses also provided a powerful way to detect strong major loci affecting monogenic and oligogenic cases.

Variable trait complexity spectrum is mainly caused by large effect variants leading to extreme phenotypes. Interestingly, we found that these causal variants have very low minor

allele frequency in natural populations and are present in individuals with extreme phenotypes. This observation is in agreement with the fact that most QTNs (Quantitative Trait Nucleotides) identified in yeast via linkage mapping are low-frequency alleles [34]. In addition, the effect of rare and low-frequency variants on phenotypic variance has also been tested in the *S. cerevisiae* yeast model. This is highly relevant as a bias towards such variants was observed in a large population of 1,011 *S. cerevisiae* natural isolates, with over 90% of SNPs having a minor allele frequency (MAF) below 0.05 [24]. Independent surveys have shown that rare and low-frequency variants contribute disproportionately to growth in a wide variety of conditions as well as gene expression variation in natural yeast populations [17,19,35].

We observed that the variability of the complexity spectrum largely depends on the growth conditions. Some conditions exhibit more monogenic and oligogenic cases, and, therefore, have higher variability, as shown by Shannon's entropy. This observation suggests that the environment plays an important role in defining the complexity spectrum of growth as suggested by previous studies [36–38]. Two obvious cases of this environmental impact are the growth in the presence of $CuSO_4$ and galactose. In the presence of $CuSO_4$, growth is primarily controlled by copy number variation of the *CUP1* gene, which in many genetic backgrounds makes growth a monogenic trait. Similarly, when galactose is the only available carbon source, many genetic backgrounds carry deleterious variants that significantly decrease growth, again making growth a low complexity trait on many occasions. More specifically, variants in the NPA03.1 (*GAL3* G154C), YJM627 (*GAL3* C456A) and DBVPG1058 (*GAL2* A655C) parents render the trait monogenic while the *GAL3* allele of YJM421 renders the trait to be oligogenic on many occasions.

Based on our results, we can state that expressivity is pervasive, as seen for variants with strong phenotypic effect. This observation calls into question the existence of monogenic traits at the population level. Indeed, Mendelian inheritance appears to be primarily cross/trait specific rather than a simple trait-related pattern. This is probably due to the intricacies of genetic interactions and metabolic pathways combined with the extensive genetic variation, which yields a large number of allelic combinations with potential epistatic effects. This would in some cases expand the phenotypic and complexity landscape of a trait.

Altogether, our work lays the ground for a more complete and in detail exploration of variants displaying different degree of expression by testing their effects in a wider number of genetic backgrounds. However, the dynamic nature of trait complexity also raises the point that obtaining strong phenotype predictive power based on genotype alone is highly unlikely, even for traits thought to be monogenic.

## Materials and methods

### Selection of the parental isolates

The parental isolates were selected from a set of stable haploid strains produced by replacing the *HO* locus of natural isolates from the 1,011 isolates collection with resistance cassettes [17]. The *MATa* version of each isolate carries a *KanMX* cassette and the *MATα* carries a *NatMX* cassette. To capture as much of the genetic diversity of the species as possible we selected 20 haploid strains with high genetic diversity, coming from all over the globe and from many different ecological niches (S1 Table). The average nucleotide diversity between two parents is 0.56% and the highest nucleotide diversity 1.05%. Two of the selected haploids are the lab strains, the reference strain S288C and Σ1278b.

### Generating the segregant population

The 40 parental strains, 2 mating types for 20 different strains, were isolated on solid YPD media (1% yeast extract, 2% peptone, 2% glucose, 2% agar). The parental strains were then

crossed in a pairwise manner and arrayed on a plate of solid YPD media. After 24 hours at 30˚C, they were transferred to solid YPD media containing Nourseothricin (200μg/L) and G418 (200μg/L) using the ROTOR replicating robot (Singer Instruments) and incubated for 24h in order to select for hybrid cells. To induce sporulation, the hybrids were replicated on sporulation media (2% potassium acetate, 2% agar) on which they were incubated for 48 hours at 30˚C.

## Tetrad dissections

For each cross we carried out tetrad dissections in order to obtain 40 tetrads with 4 viable spores. The ascus walls were digested by suspending the cells in a 0.1mg/mL solution of Zymo-lyase (MP Biochemicals, Zymolyase 20T) for 15 minutes. Digested asci were then dissected using the SporePlay <dissection microscope (Singer Instruments). The spores coming from those fully viable tetrads along with the parents of each cross were then arrayed in 384-well plates with liquid YPD media and were then stored at -80˚C.

## Phenotyping/Growth screening

Colony growth was assessed on solid SC media (SC Yeast Nitrogen Base with ammonium sulfate 6.7 g/l, amino acid mixture 2 g/l, agar 20 g/l, glucose 20 g/l) supplemented with various compounds (S3 Table). Segregants were incubated for 24 hours on SC media in matrices of 1,536 density format; each segregant was present in duplicate on the same plate. Colony size was captured both before and after incubation by taking a photo of each plate with a phenotyping platform that was developed in-house. The size of each colony was then quantified in R using the gitter package [39]. All colonies with an endpoint colony size ($size_{24h}$) under 200 on the control condition (SC) were removed from the dataset due to insufficient baseline growth (n = 282,854, 11.11%). Initial colony size ($size_{0h}$) was subtracted from the endpoint colony size ($size_{24h}$) to infer colony growth during the 24-hour incubation. If the final value was negative, it was manually reassigned to zero (n = 37,494, 1.65% of all colonies). Then, we calculated the ratio between colony growth on media containing the compound ($colony\ growth_{condA}$) and on a reference condition without any compound supplementation ($colony\ growth_{SC\_ref}$).

$$colony\ growth = size_{24h} - size_{0h}$$

$$\text{growth ratio} = \frac{colony\ growth_{condA}}{colony\ growth_{SC\_ref}}$$

Finally, we combined the growth ratios of the duplicates of each individual to calculate its mean growth ratio. The correlation between duplicates of the same segregant under the same conditions is high (R = 0.96, p-val < 2.2e-16) (S7 Fig). As a result, we obtained a total of more than 1 million phenotypes.

## Phenotype simulations

To form simulated phenotypic distributions, we generated simulated phenotypes for 160 individuals grouped in 40 tetrads. For each distribution the trait is controlled by a primary "gene" and a set of 100 secondary "genetic variants". The effect/weight of the primary "gene" ($w_1$) on the phenotype is manually assigned from 0 to 1 while the effects of the secondary "genes" ($w_{2\text{-}101}$) are randomly distributed values whose sum is 1- $w_1$. To assign the presence or absence of the "genes" in each individual we carried out a simplified simulation of meiotic segregation. In this simulated segregation we combined a string of 101 ones (presence) with a string of 101

zeros (absence) at 2 breakpoints in a two-step fashion to simulate meiotic recombination. Each "gene" was randomly assigned to a position on the strings produced by the simulated meiosis and the phenotypic value of the individual is the sum of the products between the presence of the variants and their weight. To simulate phenotype controlled by recessive epistatic interactions we carried out simulations with 2 primary genes with a common $w_1$ where both primary "genes" would have to have a state of 1 in order for them to influence the phenotype. To simulate dominant epistatic interactions, if only one of the 2 primary "genes" had a state of 1 their influence on the phenotype would take place.

Finally, we introduced an experimental noise component to our phenotypes by adding a random value from a normal distribution with a mean of 0 and a variance equal to the variance of replicates of the same individual. In total we generated 25,000 non-epistatic, 12,500 recessive epistatic and 12,500 dominant epistatic distributions.

## Classification of distributions

The first step in classifying the phenotypic distributions of the segregants was to differentiate bimodal from unimodal distributions. For this task, we used the 50,000 simulated phenotypic distributions mentioned above to train a random forest model to infer the weight of the main variants by assessing the shape of the distribution and the segregation of the phenotypes in the tetrads. The weight of the main variant is later used as a proxy for the bimodality of the phenotypic distribution because it is the main factor determining the bimodality of the distribution.

In total, a set of 24 features calculated from the phenotypic distributions were used for the random forest. The first 11 features were the quantiles of the growth ratio for every 10% of the population. Using the *emtest.norm* command of the MixtureInf package, we fitted a mixture model of two normal distributions to the simulated phenotypic distributions [40]. We used the means and standard deviations of each of the two normal distributions as well as the t-statistic of emtest.norm. We also calculated the ashman statistic [41] from the mean and standard deviation of the mixture model. The score and p-value of a Kolmogorov-Smirnov test comparing the phenotypic distribution to a normal distribution of equal mean and variance, were also used as features. Finally, the segregation of the phenotypes in the tetrads, captured by the frequency of tetrad types with 0, 1, 2, 3 or 4 segregants above average phenotype, were the last 5 features of the model.

The random forest of 100 trees was created using the randomForest package [42] and trained with the 50,000 simulated phenotypic distributions to predict the level of weight of the main variant that is used as a proxy for the bimodality of the distribution. To evaluate the algorithm, we manually annotated the bimodality of 545 phenotypic distributions from our screening experiment and predicted their bimodality using the model that was compared to the manual annotations. We established an ROC (Receiver Operator Characteristic) curve and selected a threshold value that was an equal compromise between specificity and sensitivity; the area under the ROC curve (AUC) is 0.977, which shows that the model is reliable. The random forest algorithm was then used to predict if the phenotypic distributions obtained during the screening were bimodal or unimodal. The cross/trait combinations having unimodal distributions were assigned as cases of complex traits. To assign complexity levels to the cases having bimodal distributions, we used a decision tree that considers the ratios of the two modes of the distribution and the position of the parental phenotypes in respect to the two modes. We fit a mixture model of two normal distributions (using the package flexmix) and calculated the positions of the two parental strains in respect to the 2 normal distributions [43]. We also used the proportion of segregants belonging to each of the normal distributions provided by the mixture model. In the cases where the mixture model couldn't fit, we

calculated the derivative of the distribution to locate the peak of each of the groups of spores as well as the minimum between them. The minimum point was assigned as the threshold separating the two groups of spores, from which we calculated the positioning of the parental strains as well as the proportions of the two groups of segregants. All the data processing and analysis was done using R.

### Entropy as a proxy for expressivity

The Shannon entropy of genetic complexity was calculated across the crosses of each parent using the *Entropy* function of the *DescTools* library in R with the default parameters [44].

### Bulk segregant analysis

For each of the selected crosses we picked 100 segregants with low growth in their respective condition and 100 spores with high growth to form 2 pools. Genomic DNA was extracted from each pool using the MasterPure YeastDNA purification kit (Epicentre) using the manufacturer's protocol and was sequenced by MiSeq 75bp paired end sequencing. The sequences of each pool were then aligned to the genome of one of the parents of its cross using BWA [45] and the variants were called with the command HaplotypeCaller of GATK [46]. The vcf file was imported into R with the vcfR package [47] and the allele frequency plots were made with ggplot [48].

### Phenotype rescue by plasmid insertion

The low growth parents were transformed with a centromeric plasmid containing the S288c allele of the candidate gene [32]. We then screened both the transformed parent and the non-transformed parent (negative control) on the condition of interest in the same conditions as the screening of the segregant population.

### Exploring the effects of variants in the natural population

For each of the crosses analyzed by bulk segregant analysis, we compared the sequences of the two parents for the candidate gene using BLAST [49] and inferred the variants between them. Using the genotypes from [24], we identified which isolates in the 1,011 collection carry each version of the variant. We compared the growth phenotypes of the natural isolates with the different versions of the variant and calculated the p-value between the two groups using a one-sided t-test. We performed Bonferroni normalization when multiple SNPs were present in the same gene. We considered corrected p-values under 0.05 as significant.

### Deleterious variants exploration

The deleterious variants in the *GAL* genes were selected by filtering the deleterious variants annotation of [24] with the positions of the *GAL* genes. We then filtered the vcf matrix of [24], using vcfR [47], to identify the most deleterious variant in each isolate. In total, 974 isolates were screened for growth on galactose 2% with the same workflow as the segregants of the diallel panel and compared the growth of isolates depending on the class of their most deleterious variant.

### Supporting information

**S1 Fig. Classification of the genetic complexity.** Results of the random forest, the proportions of the groups of spores and the phenotypes of the parents were used as the input for a decision tree that classifies the phenotypic distribution in one of 8 types of genetic complexity that were

then merged into 3 complexity levels (monogenic, oligogenic and complex).
(TIF)

**S2 Fig. Growth in galactose 2%.** Comparison between the growth ratios of the 4 parental strains producing many monogenic and oligogenic cases and the 16 remaining parental strains.
(TIF)

**S3 Fig. Bulk-segregant analysis results for the cross between YJM627 and YPS141.** Frequency of the YPS141 alleles in the sequencing reads of the segregants with high growth on galactose 2%. Important deviation towards the YPS141 alleles is observed in chromosome 4 (400,000–500,000).
(TIF)

**S4 Fig. Phenotypic distribution of the cross between YJM627 and NPA03.1.** Distribution of the growth phenotypes on galactose 2% of the segregants from the cross of YJM627 and NPA03.1. All segregants display low growth indicating that the large effect loci in the two parents are under genetic linkage and therefore positioned in the same region.
(TIF)

**S5 Fig. Bulk-segregant analysis results for the cross between YJM627 and DBVPG1058.** Frequency of DBVPG1058 alleles in the sequencing reads of the segregants with low growth on galactose 2%. Important deviations are observed in two regions. On chromosome 4, there is a decrease of DBVPG1058 allele frequency, suggesting a deviation toward YJM627. On chromosome 12, there is a significant increase of the DBVPG1058 allele frequency.
(TIF)

**S6 Fig. Bulk-segregant analysis results for the low growth segregants of the YJM421 and HN16 cross Frequency of the HN16 alleles in the sequencing reads of the segregants with low growth on galactose 2%.** Two regions, on chromosome 4 (350,000–500,000) and chromosome 5 (400,000–500,000), show important deviations towards the HN16 and YJM421 alleles, respectively.
(TIF)

**S7 Fig. Growth phenotypes.** Correlation density between replicates of the same segregant on the same condition (R = 0.96, p-val< 2.2e-16).
(TIF)

**S1 Table. Geographic and ecological origins of the parental isolates.**
(XLSX)

**S2 Table. Nucleotide divergence between the parental strains.**
(XLSX)

**S3 Table. Conditions used in this study.**
(XLSX)

**S4 Table. Levels of complexity of the crosses tested in this study.**
(XLSX)

**S5 Table. Shannon entropy values for each isolate and each condition.**
(XLSX)

**S6 Table. Large effect variants identified and associated minor allele frequency (MAF).**
(XLSX)

## Author Contributions

**Conceptualization:** Jing Hou, Joseph Schacherer.

**Data curation:** Andreas Tsouris.

**Formal analysis:** Andreas Tsouris, Téo Fournier, Anne Friedrich, Jing Hou, Maitreya J. Dunham, Joseph Schacherer.

**Funding acquisition:** Joseph Schacherer.

**Investigation:** Andreas Tsouris, Téo Fournier, Anne Friedrich, Jing Hou, Maitreya J. Dunham, Joseph Schacherer.

**Writing – original draft:** Andreas Tsouris, Joseph Schacherer.

**Writing – review & editing:** Joseph Schacherer.

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
