## [Decision Letter · Decision Letter 0]

18 Oct 2023

Dear Dr Schacherer,

Thank you very much for submitting your Research Article entitled 'Species-wide survey of the expressivity and complexity spectrum of traits in yeast' to PLOS Genetics.

The manuscript was fully evaluated at the editorial level and by independent peer reviewers. The reviewers are generally enthusiastic, but they identified some concerns that we ask you address in a revised manuscript. In particular, you should (1) provide more detail about the experimental approach (2) clarify the discussion of monogenic/complex traits (3) include more justification of the advantages of the "many by many" approach and (4) include all supplementary material.

We therefore ask you to modify the manuscript according to the review recommendations. Your revisions should address the specific points made by each reviewer.

Yours sincerely,

Geraldine Butler

Section Editor

PLOS Genetics

Geraldine Butler

Section Editor

PLOS Genetics

Reviewer's Responses to Questions

**Comments to the Authors:**

Reviewer #1: The manuscript “Species-wide survey of the expressivity and complexity spectrum of traits in yeast” by Tsouris et al explored the dichotomy between monogenic and complex traits by conducting comprehensive diallel crosses of 20 distinct parental genotypes to each other, under 38 different environmental conditions and measured growth ability as the phenotype readout. They also followed up with two conditions CuSO4 and galactose 2% that showed strong Mendelian inheritance and identified their genetic variants. Overall, the authors conducted systematic and comprehensive experimental crosses to explore the genetic architecture of traits in yeast, but there are a few major drawbacks the current manuscript faces.

1. I cannot agree with the introduction that the authors set in which there exists a dichotomy between monogenic and complex traits. This is a misconception not shared by the majority of the field. There is no theoretical conflict between Mendelian inheritance and complex traits. Consider the simplest Mendelian phenotype, determined by one gene A, a, the gametes will display two phenotypes, determined by A, a allele alone. If two genes A, B determine one phenotype, as we learned from basic Genetics, the possible gametes are AB, Ab, aB, and ab. Depending on whether there are dominance and epistasis, as well as the effect size, the resulting phenotypes of the four genotypes could differ. Even under the simplest additive assumption, if A has a large effect of 1 (a of 0), and B has a small effect of 0.1 (b of 0), the distribution of the four gametes will look bimodal. Quantitative genetics expands this theory to many genes, and under additive assumptions, the resulting phenotypes will display as a single normal distribution. The authors can refer to quantitative genetics literature for more background (e.g. PMID: 27812106). Maybe what the authors really mean is that for traits that are seemingly Mendelian, their genetic architecture can be more complicated underneath the surface, which was the main question explored by Hou et al. Cell Reports 2016. The assumptions they made in the model simulations with one major allele contributing to the phenotype while many other variants can modify the phenotype to a small degree seem to support this hypothesis. However, the authors need to rewrite their introduction to make the point clear.

2. The experimental setup in this manuscript setup is similar to a previous publication from the same group (Hou et al. Cell Reports 2016), with the difference being this is a many by many crossing scheme, than the previous one to many scheme. The authors claim that there could be a bias of the previous one to many scheme. However, despite of their efforts, their main findings don’t seem to differ much from previous results. For example, they found a 4.4% Monogenic and 9.6% Oligogenic in this study compared with 8.9% from previous study, with the majority traits being complex. They also observed similar results for the growth under CuSO4 – around half displayed Mendelian bimodal distribution and half being complex. They also identified the same CUP1 gene, although this manuscript found the relationship of the copy number of CUP1 gene to the distinct growth patterns. They did develop a different computational method to train and predict the types of phenotype, and performed bulk segregant analysis to follow up with the underlying genetic architecture in galactose. If the authors can predict the number of genes underlying those phenotypes for complex traits, it will be one step advance. Though adding new genotype combinations may not change the phenotypic distribution much (Figure 2C, most genotype parents behave similarly under each environment), I think this study is still helpful to get a more comprehensive view of the genetic architecture of complex traits. This result suggests that the environment could play a larger role than genotype diversity in affecting phenotypes. This has been directly or indirectly suggested by previous QTL-type or variant library studies (Nguyen Ba et al. eLife 2020, Jackson et al. eLife 2020). The authors need to expand the discussions on this point to make it a larger impact.

I also have a few minor points noted:

a) Oligogenic may not be familiar with many audiences. The authors need to say the definition and how they define such group in their calculations. I only found a supplementary figure on this, but I think the authors should make it clear in the main text and the methods.

b) The authors use growth rate as the phenotype, but they mention fitness distribution directly. It can be quite confusing since the term fitness is used in other context to calculate selection coefficient to evolutionists (page 4, the fifth to last sentence).

c) For their computational model, they only briefly mentioned that they are trained on simulations in the main text. In the methods, they said the AUC is 0.977, which is pretty good number. I am a little confused about the training and test sets used with the simulations. Are the training and test set the same, or the test sets are newly generated? Are there any cross validations?

d) The author did not cite Hou et al. Cell Reports 2016 when they talk about CuSO4 results, which they should.

e) For the causal variants identified under galactose 2% in YJM627, NPA03.1, and DBVPG1058, it will be more intuitive to include a graph showing SNPs at each of the locations they identified as variants. I am curious which SNPs are linked, or can be present in other genotypes compared with a WT lab strain background.

Reviewer #2: Tsouris et al. Species-wide survey of the expressivity and complexity spectrum of traits in yeast

In study, Tsouris et al. generate a large diallel panel of Saccharomyces cerevisiae hybrids in a many by many parents scheme and characterize growth of the offspring under different conditions. The take-home message from analyzing this impressive data is that a vast majority of traits are complex, but dynamic in the sense that the spectrum of trait complexity is related to the parental background and the growth condition. Authors then describe several cases of parents-condition combinations, in the context of what is observed at the wider population level. This paper has great potential to attract attention from people in and outside the field, since it addresses one of the most fundamental open questions in genetics and contributes to it by describing the general patterns of trait-complexity distributions and providing a unique resource for future studies. I do have a couple of concerns and some suggestions:

1. The main conclusion of this work is inferred from the distribution of phenotypes and their classification, which is done in an unsupervised manner using a trained model. However, the approach is in a way taken for granted, not mentioned in eg. abstract or introduction summary and, more importantly, no test of how well this approach performs is provided. While this is described in the methods and supplementary Fig2S (schematic), I was expecting to see the actual phenotypic data and its classification in a more clear manner, at least for few examples and as part of a main ms figure. Panel B in Fig. S2 is confusing, as it combines the schematic of the approach with the actual ROC evaluation. It was also unclear to me how experimental variation is taken into account, for example to tell parental phenotypes apart from non-parental. If arbitrary or statistical cutoffs are used, one would like to see how sensitive the classification spectra are to those cutoffs.

As I side note, I suggest but giving the observed ranges and not the 86/9.6/4.4% frequencies throughout the ms (abstract, introduction summary, discussion recap). Results in Fig2B are more revealing than those in Fig2A, what is currently being highlighted.

2. The actual phenotypic data (growth ratios) are not shown or further discussed. Showing this at least as supplementary could serve many purposes, such as describing the general patterns of segregation, illustrating how complexity is classified, or letting the reader get a better sense of how impressive the data set is. I assume that there is strong variation in the correlation of growth ratios, both across strain crosses and conditions. Are growth ratios more correlated within the categories and sub-categories defined in Table S2 or are there already unexpected behaviors by just looking at such correlations? How common are transgressive phenotypes and is this a general trend or also condition-specific?

Growth data should be made available in the supplementary material.

3. In my opinion, this study is a significant step forward in the context of previous research by the authors (Hou et al 2016) in which one single strain was crossed to many. But a stronger case needs to be made in this manuscript, in terms of analyzing and discussing the results presented here. In the introduction, potential limitations of using the one-by-many approach are indeed put forward but are later missing in the context of their results. What had we missed previously from only looking at sigma1278 or any single strain as a fixed parent? Is it possible to predict how much are we still missing when looking at 20? In what sense was the classification strategy different in the new study? The available data should be used to address this or similar questions in results and later in the discussion.

4. I think one new aspect of the current study is the systematic description of the so called oligogenic traits. A specific case is nicely described while looking at the genetic underpinnings of growth in galactose. However, it was unclear to me how robust is the overall classification of these traits. I guess part of the answer will come from addressing comment#1 above. But one would then naturally wonder whether the dataset and method allow for scoring the more specific examples involving different types of genetic interactions generated in the simulated phenotypic dataset. Is it possible at least to show what the most common sub-type of interaction is inferred in the oligogenic traits?

5. With the current decision tree scheme, experimental data from additive allele pairs would be scored as “complex” showing unimodal distributions (multi modal only if resolution and parental differences allow for it). Perhaps I am missing something, but I would not call them complex traits, at least in comparison to the so-called oligogenic traits which are in my mind more complex than two additive alleles because not only two genes are involved but they also modify each other. Given that additive interactions are more common that modifier interactions, authors should discuss whether the “complex” trait class is being overestimated in some or all cases and what are the possible alternatives to addressing this issue.

6. Part of the supplementary material was very hard to follow, in part because of trivial flaws such as incorrect labeling, missing info, inconsistent referencing in text, and so on. I am listing a couple of examples below, but these are not exhaustive. Authors should take care of this major issue throughout.

These are other comments and suggestions:

-The genetic basis of CuSO4 and growth in galactose are elegantly addressed in the last two sections of the manuscript, but this interesting part of the study is not featured in the abstract or recapitulated later in the discussion in a more explicit manner.

-Methods: Growth rations were scored in duplicate and later combined. Please show their overall correlation in a scatter plot to assess if averages are sound, the fraction of outliers, and whether some conditions were noisier than others.

-“Epistasis” should not be part of the keywords, at least in the current version of the ms.

-PG2par3: “… this study suffered from several biases”. Only one bias is suggested (one by many, with bias due to possible strong allelic effects (comment #3 above).

-Figure 1C,D: I don’t think these panels add much, specially panel D. A descriptive figure with data in Tables S1 and/or S3 would be more informative.

-PG6par1: “From more than three million measurements…”. Would be useful to see this (comment #2 above).

-I could not see supporting FigS1 cited in the text or why these comparisons are relevant. State in legend whether the 1011 data was generated here or taken from elsewhere (reference).

-Table S4: Table has two headings, S2 and S4.

-Table S4: How was each cross labeled? Could not find this in the supp material.

-Table S4: A small set of crosses/conditions are not shown (6872 in table). Show as N/A and state in text or note why they are missing.

-Figure 2C and label: Conditions are sorted by entropy. Are parents sorted some way? Perhaps should

-PG7 Add copperSO4 resistance or similar in section heading

-Figure 3B: Rank by complexity, as in Fig2C. Same comment for Fig4A.

-While going through the galactose complexity section, it was hard to follow results in Figure4, supplementary figures and tables. Authors should check if there is a better choice of main figure panels or at least try to be consistent in the order in which supp figures appear in text.

-There are several errors in some supp figures and labels: eg. ylabel in FigS6, legend in FigS5 has nothing to do with what I see in the fig. I did not check thoroughly, but it is likely there are more mistakes like this.

-Some sup figures are not cited in text or seem to be incorrectly cited in text: references seem to be missing at least for supp Figure S1, Table S6, S7.

-Some supporting tables need more descriptive titles or notes.

-Table S6: What is going on with methyl viologen and FRE1? I did not see this described in the main text.

-Fig4B and label: better to use the same color codes (blue tones) as in Fig4A and other figures in the ms.

-PG12par2: “The last parent leading to… (Figure 4B). I think Fig4A better shows this.

-I think Figure 4D is not cited in text.

-Figure 5: Show n (number of natural isolates) in figure or label. Or show the actual data points. Will help making the point that large effect variants are less common.

-PG15par1: Also give the full range of nucleotide diversity. From table S3 it seems to range from about 0.1 to 1%

-PG15par3: “to obtain *40 tetrads”, right? (160 segregants, right?)

-PG16par1: “All colonies with an endpoint..” Say how many and %

“If the final value was negative, …” Say how many and %

-PG15, Growth screening: Perhaps it is just that I am not familiar with the experimental setup, but it was unclear to me why measuring the initial colony size is important. Is initial cell density enough to be scored or would it be better to screen initial sizes few hours after? Is this relevant because of technical inoculum variation or density of the saturated cultures? Also, not clear why 24h is used, I guess to maximize differences before saturation. Was this at all calibrated here? Or provide the relevant reference, if done previously.

-PG18, Phenotype rescue: It was not clear to me if the phenotyping setup and data analysis was the same for these lower-scale experiments than the large-scale screening. Also 24h? Also duplicates? This seems to be Table S7, but there is no information in the table nor is it cited in text.

Reviewer #3: Thank you for the opportunity to review “Species-wide survey of the expressivity and complexity spectrum of traits in yeas” by Tsouris et al. This manuscript describes the spectrum of trait complexity in yeast. Authors conduct 190 crosses between 20 natural isolates capturing the genetic diversity of S. cerevisiae. They subject each cross to 40 tetrad analyses and assess the growth of the resulting meiotic progeny on 38 different conditions and analyze the resulting colony sizes. They also conduct bulk segregant analyses to identify causal loci. They generate a random forest model to classify segregation patterns into monogenic, oligogenic and complex. They show that most traits are complex and the trait complexity is dynamic, specific to traits and genetic background.

The idea to explore the spectrum of trait complexity in natural populations is interesting and important. Other studies examine traits as monogenic or complex and the novelty of this study is considering the full range of trait expressivity. This is an elegant study, and the manuscript is well-written and logical. I only have minor suggestions that the authors may wish to consider in a revised manuscript.

Minor comments

1. Authors measure entropy in Figure 2C but this result is never mentioned in the discussion. Authors should elaborate on the significance of this finding and place it in context of other findings.

2. Figures 3B and 4A contain a typo in the heading 'complextiy' should be 'complexity'.

3. On page 10 authors cite Ho et al 2009 for using the centromeric Moby ORF plasmid library and then discuss the variation in S288C ORF that rescued the growth defects. However, the Moby ORF library also contained native promoter and terminator sequences: ~900 bp upstream of the start codon and ~250 bp downstream of the stop codon. Authors should discuss how this influences their findings.

4. In general, the discussion is very short. Authors should also put into context the specific examples that they highlight in the results section i.e. CUP1, GAL3, GAL2 and COX15.

**Have all data underlying the figures and results presented in the manuscript been provided?**

Reviewer #1: **No: **The authors did not include raw growth rate measurements. The authors also did not include the raw sequencing files for bulk sequence analysis.

Reviewer #2: **No: **raw phenotypic data is missing

Reviewer #3: Yes

PLOS authors have the option to publish the peer review history of their article (what does this mean?). If published, this will include your full peer review and any attached files.

Reviewer #1: No

Reviewer #2: No

Reviewer #3: No

---

## [Editor Report · Decision Letter 1]

2 Jan 2024

Dear Dr Schacherer,

We are pleased to inform you that your manuscript entitled "Species-wide survey of the expressivity and complexity spectrum of traits in yeast" has been editorially accepted for publication in PLOS Genetics. Congratulations!

Yours sincerely,

Geraldine Butler

Section Editor

PLOS Genetics

Comments from the reviewers (if applicable):

**Data Deposition**

http://datadryad.org/submit?journalID=pgenetics&manu=PGENETICS-D-23-00959R1

**Press Queries**

---

## [Editor Report · Acceptance letter]

11 Jan 2024

PGENETICS-D-23-00959R1 

Species-wide survey of the expressivity and complexity spectrum of traits in yeast 

Dear Dr Schacherer, 

We are pleased to inform you that your manuscript entitled "Species-wide survey of the expressivity and complexity spectrum of traits in yeast" has been formally accepted for publication in PLOS Genetics! Your manuscript is now with our production department and you will be notified of the publication date in due course.

With kind regards,

Zsofia Freund

PLOS Genetics

On behalf of:
